# Non-antibiotic medication use in an Indonesian community cohort 0–18 months of age

Jarir At Thobari [1,2,3] *, Cahya Dewi Satria[2,3], Yohanes Ridora[2], Emma Watts[4], Amanda Handley[4,5], Jane Standish[4,6], Novilia S. Bachtiar[7], Jim P. Buttery[4,8,9,10,11], Yati Soenarto[2,3], Julie E. Bines [4,8,12]

1 Department of Pharmacology and Therapy, Faculty of Medicine, Public Health and Nursing, Universitas Gadjah Mada, Yogyakarta, Special Region of Yogyakarta, Indonesia, 2 Pediatric Research Office, Faculty of Medicine, Public Health and Nursing, Universitas Gadjah Mada, Yogyakarta, Special Region of Yogyakarta, Indonesia, 3 Department of Paediatrics, Faculty of Medicine, Public Health and Nursing, Universitas Gadjah Mada, Yogyakarta, Special Region of Yogyakarta, Indonesia, 4 RV3 Rotavirus Vaccine Program, Murdoch Children's Research Institute, Parkville, Victoria, Australia, 5 Medicines Development for Global Health, Melbourne, Victoria, Australia, 6 Department of General Medicine, Royal Children's Hospital Melbourne, Parkville, Victoria, Australia, 7 PT Bio Farma, Bandung, West Java, Indonesia, 8 Department of Pediatrics, University of Melbourne, Melbourne, Victoria, Australia, 9 Department of Paediatrics, Monash University, Clayton, Victoria, Australia, 10 School of Public Health and Preventive Medicine, Monash University, Clayton, Victoria, Australia, 11 Department of Infection and Immunity, Monash Children's Hospital, Clayton, Victoria, Australia, 12 Department of Gastroenterology and Clinical Nutrition, Royal Children's Hospital Melbourne, Parkville, Victoria, Australia

* j.atthobari@ugm.ac.id

**Data Availability Statement:** All relevant data are within the paper and its Supporting Information files.

## Abstract

### Background

Rational medication use for treatment is mandatory, particularly in children as they are vulnerable to possible hazards of drugs. Understanding the medication use pattern is of importance to identify the problems of drug therapy and to improve the appropriate use of medication among this population.

### Methods

A post-hoc study of the RV3-BB Phase IIb trial to children aged 0–18 months which was conducted in Indonesia during January 2013 to July 2016. Any concomitant medication use and health events among 1621 trial participants during the 18 months of follow-up were documented. Information on medication use included the frequency, formulation, indication, duration of usage, number of regimens, medication types, and therapeutic classes.

### Results

The majority of participants (N = 1333/1621; 82.2%) used at least one non-antibiotic medication for treatment during the 18-month observation period. A total of 7586 medication uses were recorded, mostly in oral formulation (90.5%). Of all illnesses recorded, 24.7% were treated with a single drug regimen of non-antibiotic medication. The most common therapeutic classes used were analgesics/antipyretics (30.1%), antihistamines for systemic use

**Funding:** The vaccine company PT Bio Farma provided support for this study in the form of a salary for the author NSB. The specific role of this author is articulated in the 'author contributions' section. Additionally, PT Biofarma was one of the funders of the RV3-BB Phase IIb trial. Another funding source of the RV3-BB Phase IIb trial was Murdoch Children's Research Institute. The funders had roles in the study design, decision to publish, and preparation of the manuscript.

**Competing interests:** The authors have the following interests: NSB is a paid employee of PT Biofarma. PT Biofarma is a state-owned company (owned by Indonesia government) that provides all vaccines for Indonesia National Immunization Program. Additionally, PT Biofarma was one of the non-commercial funders for the RV3-BB Phase IIB trial. No consultation was done to the funder relating to the trial. The trial was a phase IIB trial and while the RV3-BB rotavirus vaccine is not yet marketed, this vaccine is being developed with the intention for eventual usage in the national immunization program. This does not alter our adherence to PLOS ONE policies on sharing data and materials.

(17.4%), cough and cold preparations (13.5%), vitamins (8.6%), and antidiarrheals (6.6%). The main medication types used were paracetamol (29.9%), chlorpheniramine (16.8%), guaifenesin (8.9%), zinc (4.6%), and ambroxol (4.1%). Respiratory system disorder was the most common reason for medication use (51.9%), followed by gastrointestinal disorders (19.2%), pyrexia (16.9%), and skin disorders (7.0%).

## Conclusion

A large number of children were exposed to at least one medication during their early life, including those where evidence of efficacy and safety in a pediatric population is lacking. This supports the need for further research on pediatric drug therapy to improve the appropriate use of medication in this population.

## Introduction

Rational and judicious use of medicines is necessary at all ages, and critical in childhood where pre-clinical safety and efficacy data is more limited. In children, this importance of rational drug use is also compounded since they are vulnerable to possible hazards of drugs [1,2]. Several studies showed that off-label and contraindicated drug use for this age are still a major problem [3–6]. A study focusing on drug prescriptions among children at a community pharmacy located in South Jakarta, Indonesia, concluded that polypharmacy, inappropriate dose, and potential drug interactions were still a major problem in that setting [7]. Many drugs exhibit different pharmacokinetic, pharmacodynamic, efficacy and safety effects between children and adults [1,2,8,9]. Therefore, one of the solutions to improve rational use of drugs in children is by understanding the medication use profile among this population [10].

Several studies have been conducted regarding medicine use in children, including those conducted in the early 2000s [1,10–17]. Most of the studies analyzed drug prescriptions from healthcare settings, including pediatric wards, outpatient clinics, or community pharmacies. A more recent study [11] provided an overview of drug use in children in three European countries, suggesting that children less than 2 years old had the highest prescription rate in all three study countries. In Indonesia, information regarding the drug use profile in young children is insufficient. This study is a *post-hoc* analysis of a phase IIb trial of rotavirus vaccine in infants 0–18 months of age in Indonesia which analyzed all non-antibiotic medications used by the trial participants during the study. Antibiotic use in the same trial participants has been described previously [18].

## Method

### Study design

This study used secondary data from a phase IIb randomized, double-blinded, controlled trial for RV3-BB rotavirus vaccine (Australian New Zealand Clinical Trials Registry number ACTRN12612001282875; the protocol is available at NEJM.org). The RV3-BB phase IIb trial was conducted primarily to evaluate the efficacy of RV3-BB vaccine against severe rotavirus gastroenteritis in children aged up to 18 months. The complete study design of RV3-BB phase IIb trial has been described previously [19] and is briefly summarized here.

## Participants

The RV3-BB phase IIb trial was conducted from January 2013 to July 2016 in *Puskesmas*/primary health centers (PHCs) and hospitals in two provinces, Central Java and Yogyakarta, in Indonesia. The trial involved a total of 1649 participants, and 1621 of them gave consent for their information to be used in future studies (Fig 1). Only 49 of 1621 participants (3%) were not completely followed until 18 months. Some of these participants used at least one non-antibiotic medication before being lost to follow-up, therefore we still included these 49 participants in the analysis.

Twenty-three PHCs and two hospitals in Sleman district (an urban area in Yogyakarta province) and Klaten district (a rural area in Central Java province) participated as study sites. After obtaining written informed consents from participants' mothers during pregnancy, healthy neonates were then enrolled in this study up to 6 days of age. The key inclusion criteria were neonates who were healthy, born full-term, and weighed between 2500 and 4000 grams.

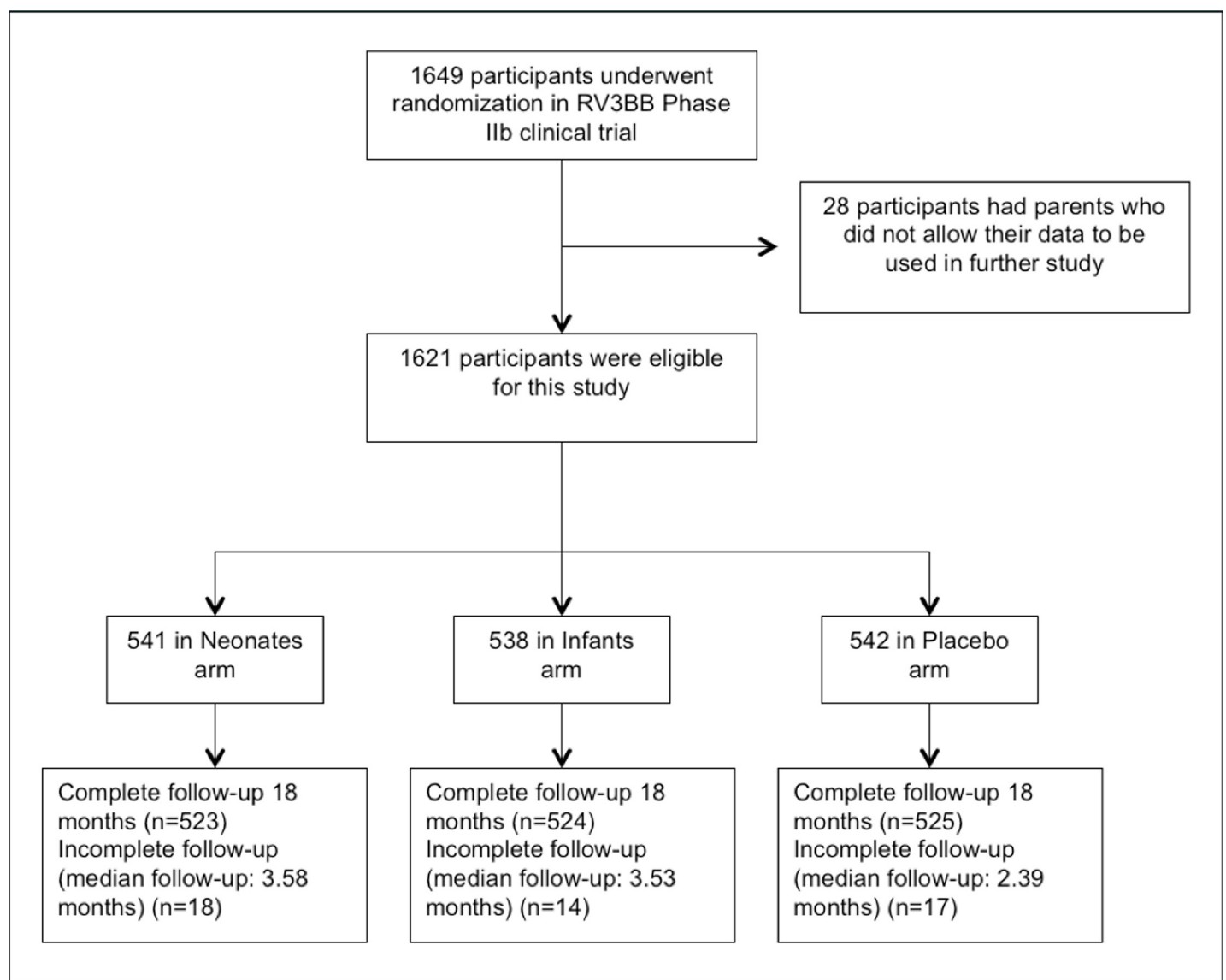

**Fig 1. Flow diagram of participant follow-up.**

## Interventions, randomization and blinding

Eligible participants were assigned to one of the three intervention groups (1:1:1): neonatal-schedule vaccine group, infant-schedule vaccine group, and placebo group. Each vaccine group received 3 doses of RV3-BB vaccine and 1 dose of placebo according to their assigned schedule, while the placebo group received 4 doses of placebo only. The 4 doses of vaccine or placebo were administered as the following schedule: dose 1 at 0–5 days of age, dose 2 at 8–10 weeks of age, dose 3 at 14–16 weeks of age, and dose 4 at 18–20 weeks of age. In neonatal schedule, the RV3-BB vaccines were administered at dose 1, 2, and 3, and followed by placebo at dose 4. In infant schedule, the placebo was administered at dose 1, and followed by RV3-BB vaccine at dose 2, 3, and 4.

The trial-group assignment was performed randomly by using a computer-generated code with a block size of 6, stratified by province. All investigators, trial monitors, data managers, other trial staff and participants' families were masked to the trial-group assignment until the end of the study. Only the pharmacist at the central pharmacy in each province who dispensed the RV3-BB vaccine or placebo was aware of the trial-group assignments.

## Sample size

The detailed sample size calculation has been described previously [19]. Briefly, our calculation generated an enrollment target of 549 participants in each of the three trial groups. This calculation was based on a power of 80% to reject the null hypothesis of no difference between the combined vaccine group and the placebo group, with an assumption of 3% of the placebo group would have a severe rotavirus gastroenteritis, and the true efficacy of the vaccine was 60% at one-sided alpha level of 0.1. With this sample size, the allowance for nonadherence rate to the trial protocol was 10%.

## Concomitant medication and reasons for the usage

During the trial, prospective surveillance of concomitant medication and adverse events (AE) was performed. Concomitant medication was defined as all medications used by the participants to treat AE, excluding antibiotics (has been published previously [18]). Drugs used to prevent an illness (prophylaxis), including vitamin K injection to all newborns after delivery and antipyretics given by the caregiver after immunization without pre-existing fever were also excluded. Medication types and therapeutic classes were classified on the basis of the WHO Anatomical Therapeutic Chemical (ATC). AE episodes were defined as all illnesses or symptoms developed during the 18 months observation period after the first dose of RV3-BB vaccine or placebo. AE episodes were classified using Medical Dictionary for Regulatory Activities (MedDRA) and further narrowed down by the researchers to 12 disorders mainly according to their affected organ systems.

Research assistants made weekly phone calls to ask the mothers about participants' conditions (including AE and concomitant medication use), and participants were visited by study midwives every month for monitoring medications used by participants during the study. Both prescribed and non-prescribed medications were taken into account in the analysis of this study. If the illness or medication use was derived from healthcare visit or hospitalization, information regarding the illness or medication was extracted from their medical record.

Collected information regarding medication use included: type of drug and its therapeutic class, duration of medication use per child, drug formulation (e.g. oral (pulveres, syrup), topical, intravenous, inhalation, rectal or intratracheal), whether the medication was from outpatient or inpatient, and treatment indication. Pulveres is an oral drug formulation containing medicines all ground together and has been used for a long time in Indonesia. Number of

non-antibiotic drug regimen, which was defined as the number of different non-antibiotic medications that were used concurrently from the same start date to treat an illness, was also recorded. Concurrent use of a non-antibiotic medication with antibiotics was considered a single drug regimen in this study. Antibiotic use in RV3-BB trial participants has been described previously [18].

## Ethical considerations

This study was approved by Medical and Health Research Ethics Committee of Faculty of Medicine Universitas Gadjah Mada–Dr Sardjito General Hospital. Written informed consent was obtained from the every child's parent or guardian for the RV3-BB phase IIb trial. The subjects included in this post-hoc analysis were those who gave consent for their data to be used in future studies.

## Statistical analysis

Statistical calculations were performed by using SPSS version 23. For descriptive data, the results were presented as mean, median, frequency and percentages. Relationships between gender and vaccination group with medication use were assessed with bivariate chi-square. A logistic regression was performed to explore the relationship between the reason of medication use, drug formulation, outpatient/inpatient setting and the age of participant when using the medication with the number of non-antibiotic drug regimen.

## Result

### Baseline characteristics

Overall, 1333 of 1621 (82.2%) trial participants used at least one non-antibiotic medication during 18-month observation period. The incidence of medication use is similar between males and females, and across intervention groups. Baseline characteristics are summarized in Table 1.

### The pattern of medication use

Following exclusion of antimicrobial exposures, a total of 7586 medications were used by 1333 children, with a mean (SD) of 5.69 (4.80) drugs/child. The most common drug route of administration was oral, and most of the medications were used in ambulatory setting (Table 2). Average use of non-antibiotic medications per AE was 1.72 (SD±0.99). Moreover, 24.7% of all AE were treated with a single drug regimen of a non-antibiotic medication (Table 3).

The most common age for our participants to be exposed to their first non-antibiotic medication was between 1–6 months old (N = 1083/1333; 81.2%), followed by 0–1 month (N = 229/1333; 17.2%), 6–12 months (N = 18/1333; 1.2%), and only 0.2% (N = 3/1333) had their first non-antibiotic medication between 12–18 months of age.

**Table 1. Baseline characteristics of study participants with non-antibiotic medication use.**

| Characteristics | All (n = 1621) | Medication use (n = 1333) | No medication use (n = 288) | *p*-value | RR |
|---|---|---|---|---|---|
| Gender (%) | | | | | |
| Male | 844 (52.1) | 699 (82.8) | 145 (17.2) | 0.52 | 1.02 (0.97–1.06) |
| Female | 777 (47.9) | 634 (81.6) | 143 (18.4) | Ref | Ref |
| Vaccination group (%) | | | | | |
| Neonates | 541 (33.4) | 450 (83.2) | 91 (16.8) | 0.88 | 1.00 (0.95–1.06) |
| Infants | 538 (33.2) | 434 (80.7) | 104 (19.3) | 0.36 | 0.97 (0.92–1.03) |
| Placebo | 542 (33.4) | 449 (82.8) | 93 (17.2) | Ref | Ref |

**Table 2. The pattern of medication use in children.**

| Pattern indicator (N = 7586) | Results |
|---|---|
| **Drug formulation (%)** | |
| Oral formulation | 6865 (90.5) |
| Pulveres | 4722 (62.3) |
| Syrup/drop | 1651 (21.8) |
| Oral solution | 149 (1.9) |
| Unknown oral | 343 (4.5) |
| Topical | 513 (6.8) |
| Skin preparation | 512 (6.8) |
| Ear preparation | 1 (0.01) |
| Other | 208 (2.7) |
| Intravenous | 85 (1.1) |
| Inhalation | 67 (0.9) |
| Rectal | 55 (0.7) |
| Intratracheal | 1 (0.01) |
| **Outpatient/inpatient medication use (%)** | |
| Outpatient | 7076 (93.3) |
| Inpatient | 510 (6.7) |
| **Duration of drug use per child (days) [median (min-max)]** | 3.67 (1–63) |

## Factors associated with the number of non-antibiotic medication used per AE

A logistic regression was done to explore the relationship between gender, patient care setting, reasons for medication use, age on medication use, and route of administrations with the number of non-antibiotic medication used per AE (Table 4). Gender was the only factor that was not significantly associated with the number of medications used. Oral route administration (RR = 3.14; 95% CI = 2.12–4.65) and having respiratory disorders, gastrointestinal disorders, and unspecified pyrexia as the reasons for medication use (RR = 3.07; 95% CI = 2.18–4.32) had the highest RRs. Using medication at an age of older than 4 months was also associated with a higher number of medications used per AE (RR = 1.57; 95% CI = 1.35–1.81), as well as in inpatient care setting (RR = 1.38; 95% CI = 1.02–1.87).

## Drug classes

The ten most commonly used therapeutic classes in this study covered 98.6% of all treated infants, and 93.1% of all medications used. The largest therapeutic class used was analgesics/antipyretics, comprising 30.1% of all medications used, and overwhelmingly as paracetamol (N = 2268/2280; 99.5%). Extremely rare metamizole and ibuprofen utilizations were noted in

**Table 3. Number of non-antibiotic drug regimen.**

| Non-antibiotic drug regimen | No. of children (N(%))* | Adverse events (N(%))* |
|---|---|---|
| Single | 1143 (85.8) | 2400 (24.7) |
| Double | 572 (42.9) | 822 (8.5) |
| Triple | 508 (38.1) | 1437 (14.8) |
| More than triple | 207 (15.5) | 906 (9.3) |

*During the trial, a child could have more than one episode of treatment and adverse event

**Table 4. Factors associated with the number of non-antibiotic medication used per AE.**

| Factors | Use of ≥3 non-antibiotic medications per AE [N (%)] | Use of ≤2 non-antibiotic medications per AE [N(%)] | Adjusted RR* (95% CI) |
|---|---|---|---|
| **Gender** | | | |
| Male (N = 2415) | 545 (22.6) | 1870 (77.4) | 1.06 (0.92–1.23) |
| Female (N = 1993) | 438 (22.0) | 1555 (78.0) | Ref |
| **Patient care setting** | | | |
| Inpatient (N = 277) | 67 (24.2) | 210 (75.8) | 1.38 (1.02–1.87) |
| Outpatient (N = 4131) | 916 (22.2) | 3215 (77.8) | Ref |
| **Reasons for medication use**** | | | |
| Respiratory disorders, gastrointestinal disorders, and unspecified pyrexia (N = 3699) | 939 (25.4) | 2760 (74.6) | 3.07 (2.18–4.32) |
| Other indications (N = 709) | 44 (6.2) | 665 (93.8) | Ref |
| **Age on medication use**** | | | |
| >4 months (N = 1919) | 534 (27.8) | 1385 (72.2) | 1.57 (1.35–1.81) |
| ≤4 months (N = 2489) | 449 (18.0) | 2040 (82.0) | Ref |
| **Route of administration** | | | |
| Oral (N = 3802) | 950 (25.0) | 2852 (75.0) | 3.14 (2.12–4.65) |
| Other routes (N = 606) | 33 (5.4) | 573 (94.6) | Ref |

**Note:** During the trial, a child could have more than one episode of adverse event. Only adverse events that treated with non-antibiotic medications are included in the table.

*Relative risk was adjusted for gender, patient care setting, reasons for medication use, age on medication use, and route of administration.

**Respiratory disorders, gastrointestinal disorders and unspecified pyrexia are the largest reasons for medication use.

**The mean age on medication use was 4.0 ± 2.7 months.

this study (N = 8/2280 and 4/2280, respectively). Other main therapeutic classes were also noted, including: antihistamines for systemic use (mainly chlorpheniramine (N = 1275/1316; 96.9%)), cough and cold preparations (guaifenesin (N = 677/1020; 66.4%); ambroxol (N = 311/1020; 30.5%)), vitamins (ascorbic acid (N = 227/652; 34.8%); vitamin B6 (N = 185/652; 28.4%)), and antidiarrheals (oral rehydration salt formulations (N = 255/502; 50.8%); probiotics (N = 130/502; 25.9%)). These results can be seen in Tables 5 and 6.

## Reasons for using medication

Of all participants, 93.6% (N = 1518/1621) experienced at least one AE during the observation period. Moreover, the first AE of each participant was commonly occurred between 1–6 months old (N = 825/1518; 54.3%), followed by 0–1 months (N = 678/1518; 44.7%), 6–12 months (N = 9/1518; 0.6%), and rarely in 12–18 months old (N = 6/1518; 0.4%).

Respiratory illness, gastrointestinal illness, pyrexia and skin disorders were the most common reasons for the participants to take non-antibiotic medication during the study (95.1%). The largest therapeutic classes used for treating respiratory system disorders were antihistamines for systemic use, cough and cold preparations, analgesics/antipyretics, vitamins, and drugs for obstructive airway. For gastrointestinal disorders, the main therapeutic classes were antidiarrheals, mineral supplements, analgesics/antipyretics, vitamins, and drugs for acid related disorders. Analgesics/antipyretics was the most commonly used medication for pyrexia (91.2%). Among medications for skin disorders, corticosteroids in dermatological preparations was the largest therapeutic class used by our participants, followed by emollients-and-protectives, antihistamines for systemic use, antifungals for dermatological use, and antiseptics-and-disinfectants. Although vitamins appeared to be used in many types of AE, they were

**Table 5. Fifteen main therapeutic classes used in participants.**

| No. | Medication classes | ATC Code | No. of drug use (%) | No. of participants | Prevalence rate (per 100 participants) |
|---|---|---|---|---|---|
| 1 | Analgesics/antipyretics | N02 | 2280 (30.1) | 1093 | 67.4 |
| 2 | Antihistamines for systemic use | R06 | 1316 (17.4) | 772 | 47.6 |
| 3 | Cough and cold preparations | R05 | 1020 (13.5) | 614 | 37.9 |
| 4 | Vitamins | A11 | 652 (8.6) | 411 | 25.4 |
| 5 | Antidiarrheals and intestinal antiinflammary agents | A07 | 502 (6.6) | 312 | 19.3 |
| 6 | Mineral supplements | A12 | 490 (6.5) | 325 | 20.1 |
| 7 | Bronchodilators | R03 | 287 (3.8) | 185 | 11.4 |
| 8 | Corticosteroids, dermatological preparations (topical agents) | D07 | 189 (2.5) | 156 | 9.6 |
| 9 | Nasal preparations | R01 | 178 (2.4) | 150 | 9.3 |
| 10 | Emollients and protectives | D02 | 145 (1.9) | 123 | 7.6 |
| 11 | Drugs for acid related disorders | A02 | 80 (1.1) | 69 | 4.3 |
| 12 | Drug for constipation | A06 | 52 (0.7) | 40 | 2.5 |
| 13 | Corticosteroids for systemic use | H02 | 46 (0.6) | 38 | 2.3 |
| 14 | Antiseptics and disinfectants (topical agents) | D08 | 41 (0.5) | 41 | 2.5 |
| 15 | Antifungals for dermatological use | D01 | 38 (0.1) | 36 | 2.2 |
| | Total | | 7586 (100) | 1621 (100) | |

*See S1 Table. All of non-antibiotic therapeutic classes used by trial participants during study period

mainly used in respiratory system disorders (N = 452/652; 69.3%). We also noted that lymphatic system disorder appeared as if it was not treated with any drugs, however, it was completely treated with antibiotic as monotherapy which was not taken into account in this study. These results were summarized in Figs 2 and 3. Moreover, all adverse events episodes recorded during the study are presented in Table 7.

**Table 6. Fifteen main medications used in participants.**

| No | Type of drug | ATC Code | No. of drug use (%) | No. of participants | Prevalence rate (per 100 participants) |
|---|---|---|---|---|---|
| 1 | Paracetamol | N02BE01 | 2268 (29.9) | 1093 | 67.4 |
| 2 | Chlorpheniramine | R06AB04 | 1275 (16.8) | 753 | 46.5 |
| 3 | Guaifenesin | R05CA03 | 677 (8.9) | 465 | 28.7 |
| 4 | Zinc | A12CB01 | 414 (4.6) | 294 | 18.1 |
| 5 | Ambroxol | R05CB06 | 311 (4.1) | 226 | 13.9 |
| 6 | Salbutamol | R03CC02 | 273 (3.6) | 181 | 11.2 |
| 7 | Oral rehydration salt formulations | A07CA | 255 (3.4) | 193 | 11.9 |
| 8 | Ascorbic acid (Vitamin C) | A11GA01 | 227 (3.0) | 179 | 11.0 |
| 9 | Vitamin B6 | A11HA02 | 185 (2.4) | 140 | 8.6 |
| 10 | Probiotics | A07FA51 | 130 (1.7) | 110 | 6.8 |
| 11 | Hydrocortisone | D07AA02 | 127 (1.7) | 112 | 6.9 |
| 12 | Dexamethasone | H02AB02 | 117 (1.5) | 97 | 6.0 |
| 13 | Calcium | A12AA05 | 79 (1.0) | 55 | 3.4 |
| 14 | Other emollients and protectives | D02AX | 73 (1.0) | 64 | 4.0 |
| 15 | Ordinary salt combinations | A02AD01 | 70 (0.9) | 62 | 3.8 |
| | Total | | 7586 (100) | 1621 (100) | |

*See S2 Table. All of non-antibiotic medications used by trial participants during study period

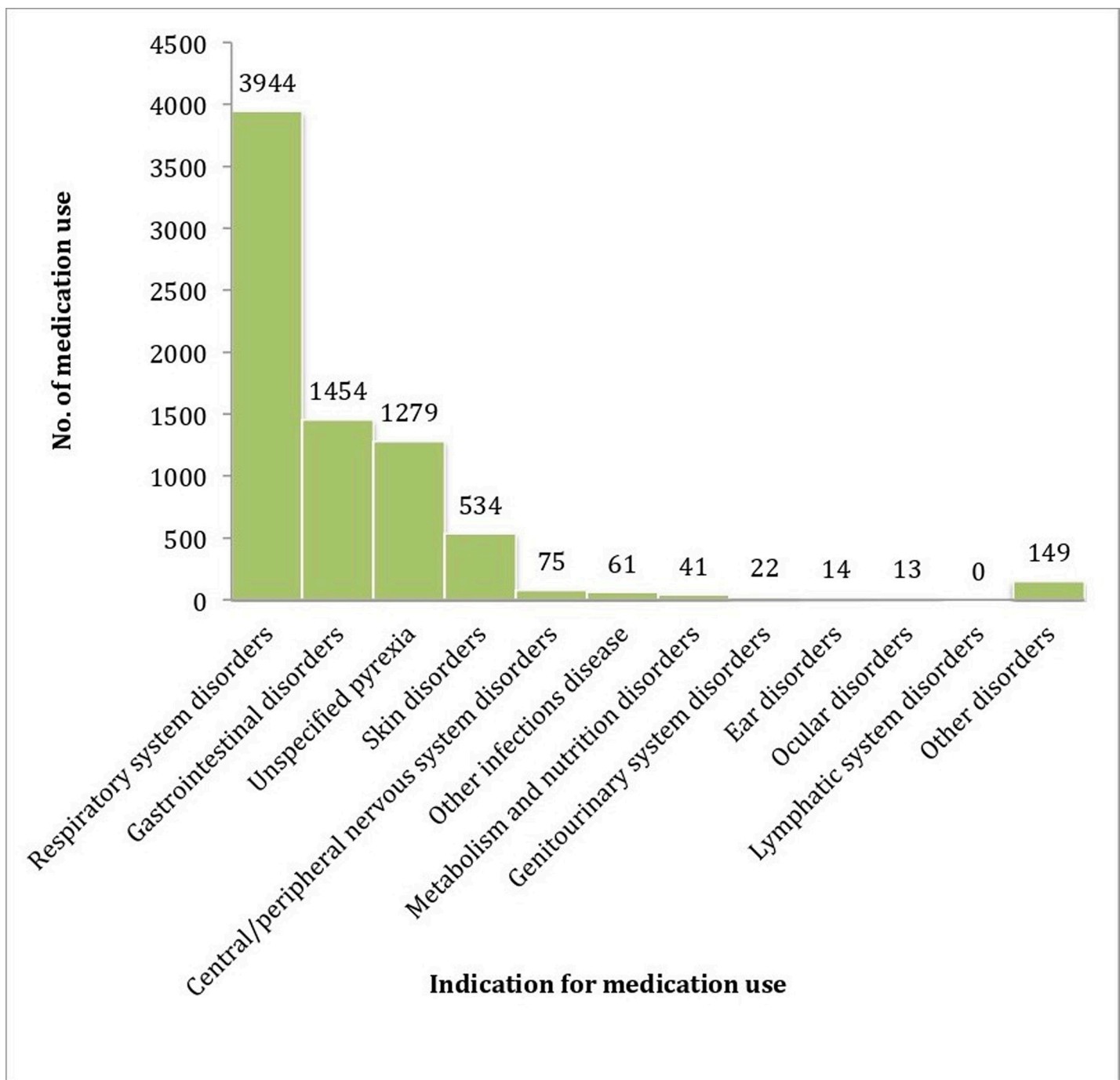

**Fig 2. Reasons for non-antibiotic medication use.** Each participant could contribute multiple times to each reason.

## Discussion

### Non-antibiotic medication use pattern

The majority of our trial participants (82.2%) used at least one medication during the 18-month of follow up. This number is comparable with similar previous studies held in Sweden [20], which suggested that: in the first year of life, 71% of the infants had at least one

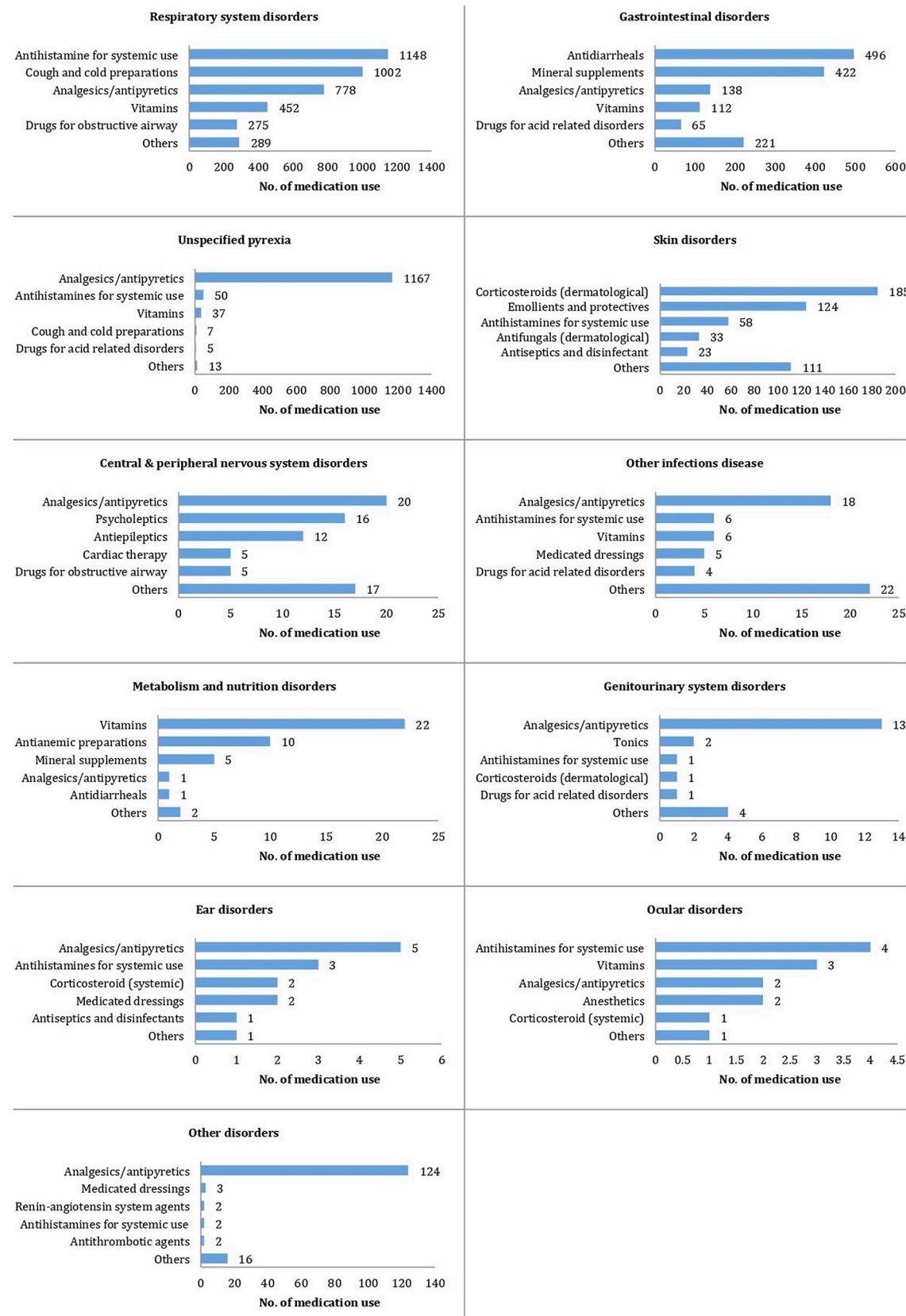

**Fig 3. Five main therapeutic classes used based on the reason for non-antibiotic medication use.** Each participant could contribute multiple times to each therapeutic class.

medication use. The number of participants using medication is similar across genders. However, other studies found different results. While a study noted that males received more drug prescriptions than females at all ages (0–14 years), some other studies showed that males only predominated at younger ages, especially in ≤10 years old, and females predominated in older children [1,17,21]. The reason for different results are difficult to determine, but most possibly due to different extent of diseases, caregiver knowledge, or local practice.

This study also did not show a reduction in the incidence of medication use in the vaccinated groups. This partly reflects that, although RV3-BB rotavirus vaccine efficacy against severe rotavirus gastroenteritis has been well established [19], it did not prevent all illnesses or medication uses. As far as we know, there have been no reports on the reduction of all medication use in children following any type of vaccinations. This is most likely due to a wide range of disease other than the illness targeted by the vaccine that could encourage children to use medication.

The most common age for our participants to be exposed to their first non-antibiotic medication was between 1–6 months old (81.2%), and followed by 0–1 month old (17.2%). A similar order of age groups was also observed in terms of the occurrence of the first adverse event. Therefore, it can be said that it is a common practice in Indonesia to expose a child at a very young age (under 6 months old) to a non-antibiotic medication once they start to get ill.

**Table 7. Adverse event episodes recorded during the study period.**

| | Adverse event | Participants (N (%)) | Total episodes of adverse event (N) | % (SE) | Episodes of AE treated by non-antibiotic medications (N) | % (SE) |
|---|---|---|---|---|---|---|
| 1 | Respiratory system disorders | 995 (65.6) | 1854 | 19.1 (0.40) | 1402 | 75.6 (1.00) |
| 2 | Unspecified pyrexia | 935 (61.6) | 1587 | 16.4 (0.38) | 1142 | 71.9 (1.13) |
| 3 | Skin disorders | 408 (26.9) | 550 | 5.7 (0.24) | 355 | 64.6 (2.04) |
| 4 | Metabolism and nutrition disorders | 29 (1.9) | 30 | 0.3 (0.06) | 12 | 40 (8.94) |
| 5 | Ear disorders | 23 (1.5) | 25 | 0.3 (0.06) | 10 | 40 (9.80) |
| 6 | Other infections | 95 (6.3) | 101 | 1.0 (0.1) | 31 | 30.7 (4.59) |
| 7 | Gastrointestinal disorders | 985 (64.9) | 2335 | 24.1 (0.43) | 697 | 29.9 (0.95) |
| 8 | Genitourinary disorders | 39 (2.6) | 43 | 0.4 (0.06) | 11 | 25.6 (6.66) |
| 9 | Central-peripheral nervous system | 55 (3.6) | 60 | 0.6 (0.08) | 15 | 25 (5.59) |
| 10 | Ocular disorders | 88 (5.8) | 95 | 0.9 (0.1) | 5 | 5.3 (2.3) |
| 11 | Lymphatic system disorders | 3 (0.2) | 3 | 0.1 (0.03) | 0 | 0 (0) |
| 12 | Other disorders | 942 (62.1) | 3026 | 31.2 (0.47) | 148 | 4.9 (0.39) |
| | Total | 1518 (100) | 9709 | 100 | 3849 | 39.6 (5) |

The table included only participants who had at least one AE during the 18 months of follow-up (N = 1518/1621). The list is ordered by the proportion of adverse event (AE) episodes treated with non-antibiotic medications. SE: Standard Error of proportion

### Formulation and duration

The predominance of oral drug formulation in our study (90.5%) was also seen in other previous similar studies [10,13,22,23]. Oral liquid formulations are widely used among young children since they are easier to swallow than the solid form [24]. However, in this study, pulveres was the most commonly used oral formulation. Pulveres is frequently used in Indonesia, especially for their lower price and flexibility to accommodate drug combinations. This may explains why oral formulation is associated with higher number of drug regimen in this study (Table 4). Considering this advantage, prescribers should be aware of any potential drug interactions when prescribing drug combination in their practice [25,26]. Rectal, inhalation, and injections were less frequently used in this study, and also in other studies [10,13,22,23]. Median duration of medication use per child in this study was 3.67 days (ranged between 1 to 63 days). However, by far, there have been no studies focusing on overall duration of medication use in infants.

### Number of and factors associated with non-antibiotic drug regimen

The average number of non-antibiotic drug regimens used was 1.72 ± 0.99 drugs per regimen. Moreover, a quarter of all AE were treated with a single drug regimen of non-antibiotic medication (Table 3). This result was lower than previous studies focusing on drug prescriptions in wider range of age population (≤12 years of age [27]; ≤5 years of age [13]) showing the average of 2.5–2.6 drugs per prescription, respectively. Fewer number of drugs per regimen in our study might be due to the younger age of population using medication which tend to have a single clinical condition, thus impact on the fewer number of medication needed [10]. This also explains a consumption of higher number of drugs by older infants (Table 4). The inclusion of prescription and non prescription drugs and the exclusion of antibiotics in our study also might affect the average number of drug regimen compared to those who calculated prescribed drugs only without excluding antibiotic, as non-prescribed drug tend to be used in milder conditions that need fewer number of drug per regimen.

Reasons for mediation use (respiratory disorders, gastrointestinal disorders, unspecified pyrexia), participant's age older than 4 months when using the medication, oral drug formulation and inpatient care setting were strongly associated with the number of non-antibiotic drug regimen. Respiratory disorders, gastrointestinal disorders, and unspecified pyrexia manifest as multiple symptoms thus may encourage child caretakers to self-medicate the children with multiple over-the-counter (OTC) [10,22]. Hospitalized child may also indicate a more complex diagnosis to be treated, thus more medications may be required [10].

### The most common therapeutic classes and the reasons for usage

The ten most commonly used therapeutic classes in this study accounted for 93.1% of all medications used by participants. A similar result was also noted in previous study which showed that the 10 most prescribed therapeutic classes in 0–1 years old children accounted for 92.7% of all drugs used [17]. Five out of those ten main therapeutic classes were similar with our study (analgesics, emollient, dermatological corticosteroid, nasal preparation, and cough and cold preparations). Similar to other studies, respiratory system disorder was the primary indication for medication use in our study [10,17,22,28].

The results of this study might represent the non-antibiotic medication use in wider population of young children in Yogyakarta and Central Java provinces in Indonesia, considering the large sample size used in the study.

**Analgesics/antipyretics.** Analgesics/antipyretics were the most commonly used therapeutic classes in this study, accounting for 30.1% of all drugs consumed, overwhelmingly as

paracetamol (99.5%), and used by 67.4% of all participants. This drug usage pattern was similar to other studies [1,10,17,23]. However, although ibuprofen and metamizole were rarely used by our participants (N = 4/7586 and N = 8/7586, respectively), both were the most frequently prescribed drugs among 0–18 years old children admitted to pediatric ward in Germany (8.9% and 8.6%, respectively) [10]. The reasons for different results were difficult to determine, but possible explanations might be due to the younger age of participants in our study, and the difference therapeutic practices, local drugs availability, or treatment guidelines [10]. Both paracetamol and ibuprofen are the recommended antipyretics in children [29,30].

**Antihistamines and cough and cold preparations.** Antihistamines and cough and cold preparations are the second and third largest therapeutic classes used in this study, with the major reason for their use as respiratory system disorders (N = 1148/1316; 87.2% and N = 1002/1020; 98.2%, respectively). These results likely reflect the fact that these are main ingredients of widely available cough and cold medications (CCMs). However, there is no good evidence of over-the-counter CCMs, such as antihistamines, decongestants, antitussives and expectorants, being superior to placebo against acute cough in young children, according to a review of pertinent trials [31] and the use of OTC CCMs in children under 6 years old were also not recommended [32–35].

Half of our participants used at least one systemic antihistamine during their first 18 month of life, which is higher compared to other studies showing 31% of 4511 children had used antihistamines by the age of 2 years in the Netherlands [1], and 17% of 1701 children during the first 5 years of life in Sweden [20]. Cough and cold preparations use in the first 18 months of life were also more frequent in our study (37.88%) than in another study in Netherlands (16%) [1]. This might partly reflect a higher incidence of respiratory tract infections in our population.

Antihistamines have been widely used for decades in clinical practice to treat hyperhistaminic conditions. However, in a meta-analysis of 35 studies focusing on antihistamine use for treating URTIs in adults and children found that there was no clinically significant effect [36]. Moreover, first-generation antihistamines (e.g. chlorpheniramine) were frequently associated with non-serious adverse effect particulary sedation, and such adverse effect might disturb patient's performance of routine tasks, particulary in older children [37].

Some studies reported child deaths associated with self-medication of CCMs, which appeared significantly higher in children under 2 years old [38–40]. An expert panel evaluating 103 children death cases involving non-prescribed CCM uses, concluded that five drugs most frequently mentioned as at least possibly related with the deaths were: pseudoephedrine (N = 43.7%), diphenhydramine (N = 36.9%), dextromethrophan (N = 35%), chlorpheniramine (N = 16.5%), and brompheniramine (N = 12.6%). There were no mentions of guaifenesin being related with the deaths according to the evaluation. The panel concluded that deaths after using these drugs were more likely from product misuse rather than their adverse effects of recommended doses [40]. Therefore, education to caregivers about the proper use of cough and cold medication in children, especially aged <2 years old, should be considered to prevent misuse of these drugs.

**Vitamins.** We noted that almost 10% of all medications used by our participants were vitamins, predominantly ascorbic acid and vitamin B6 (34.8% and 28.4%, respectively). The main reason for administering ascorbic acid and vitamin B6 was for respiratory system disorders (87.7% and 69.3%, respectively). A review of eight RCTs showed that vitamin C intake reduced the duration of URTI by 1.6 days in children [41]. However, the evidence of vitamin B6 efficacy for the treatment or prevention of respiratory system disorders was rarely studied. Possible explanations of the utilization in our study are: (1) parental or prescriber (for prescribed drugs) lack of knowledge of vitamin B6 indication; (2) local supplies of particular

vitamin that occasionally wear out thus substituted with other type of vitamin; (3) local belief that any type of vitamin would improve child's appetite which tend to be decreased due to illness [42]. Although not being investigated in children, vitamin B6 overdose in adult has been known to induce neurotoxic syndrome [43]. Therefore, the exact reason of its utilization should be further evaluated to improve the rational use of this drug in young children.

**Antidiarrheals, intestinal antiinflammary/antiinfective agents.** Gastrointestinal disorders were the second most common reason for the participants to take medication, and antidiarrheals were the largest drug class taken, followed by mineral supplements. These results were reasonable as the national and international guideline for management of childhood illnesses recommended the use of oral rehydration salt formulations and zinc as the main therapy for children presenting with diarrhea [44,45]. Probiotics, which was overwhelmingly used for treating gastrointestinal illnesses (98.5%), accounted for a quarter of all antidiarrheal classes used in this study. This is also reasonable as a Cochrane review of 63 studies (56 involved infants and young children) concluded that, probiotics reduced the duration of diarrhea by 25 hours and the risk of diarrhea lasting >4 days by 59% [46]. Moreover, the minimal use of antimotility drugs to treat diarrhea (N = 4/1200; 0.3%) in this study was judicious due to the high rates of side effects among infant and young children, including ileus, respiratory depression, and coma [47].

## Conclusion

The majority of children were exposed to at least one medication during their early life. A variety of medications were used, including those where the evidence of efficacy and safety in pediatric population is lacking. This calls for further education to improve caregivers' knowledge on appropriate self-medication in children, and to enhance the guideline compliance by health practitioners to ensure the therapy given to children will be of most benefit.

## Supporting information

**S1 Checklist. Consort checklist of post-hoc analysis of RV3-BB phase IIb trial.**
(DOCX)

**S1 Table. Therapeutic classes used by trial participants during study period.**
(DOCX)

**S2 Table. Type of drugs used by trial participants during study period.**
(DOCX)

**S1 Dataset.**
(XLSX)

## Acknowledgments

We would like to express our special gratitude towards the research team of RV3 Rotavirus Vaccine Program from Murdoch Children's Research Institute, Bio Farma, and Pediatric Research Office. We would also like to thank our colleagues from Dr. Sardjito General Hospital (Yogyakarta), Soeradji Tirtonegoro General Hospital (Klaten), Sleman General Hospital (Sleman), and participating PHCs. We also would like to show our gratitude to all project managers, research assistants, study site staff especially the midwives who have monitored the medications received by the participants, and participants' parents in Sleman and Klaten district who have been cooperating in this study. Finally, we would like to acknowledge our colleagues from Sleman and Klaten District Health Office for field research support.

## Author Contributions

**Conceptualization:** Jarir At Thobari, Cahya Dewi Satria, Novilia S. Bachtiar, Jim P. Buttery, Yati Soenarto, Julie E. Bines.

**Data curation:** Jarir At Thobari, Emma Watts, Amanda Handley, Jane Standish.

**Formal analysis:** Jarir At Thobari, Yohanes Ridora, Emma Watts, Jane Standish.

**Investigation:** Jarir At Thobari, Cahya Dewi Satria.

**Methodology:** Jarir At Thobari, Cahya Dewi Satria, Emma Watts, Amanda Handley, Jane Standish, Novilia S. Bachtiar, Jim P. Buttery, Yati Soenarto, Julie E. Bines.

**Project administration:** Jarir At Thobari, Cahya Dewi Satria, Emma Watts, Amanda Handley, Jane Standish.

**Resources:** Novilia S. Bachtiar, Jim P. Buttery, Julie E. Bines.

**Supervision:** Jim P. Buttery, Yati Soenarto, Julie E. Bines.

**Validation:** Jarir At Thobari, Emma Watts, Jane Standish, Julie E. Bines.

**Writing – original draft:** Jarir At Thobari, Yohanes Ridora, Emma Watts, Jim P. Buttery.

**Writing – review & editing:** Novilia S. Bachtiar, Jim P. Buttery, Yati Soenarto, Julie E. Bines.

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
