## [Decision Letter · Decision Letter 0]

25 Nov 2019

PONE-D-19-26757

Non-antibiotic Medication Use in an Indonesian Community Cohort 0-18 Months of Age

PLOS ONE

Dear Dr At Thobari,

Thank you for submitting your manuscript to PLOS ONE. After careful consideration, we feel that it has merit but does not fully meet PLOS ONE’s publication criteria as it currently stands. Therefore, we invite you to submit a revised version of the manuscript that addresses the points raised during the review process.

The manuscript has been assessed by two reviewers; their comments are available below. The reviewers find the work of relevance but have raised some recommendations regarding data presentation as well as comments around the statistical analyses undertaken.

Could you please revise the manuscript to carefully address the concerns raised by the reviewers?

We would appreciate receiving your revised manuscript by Jan 07 2020 11:59PM. Please include the following items when submitting your revised manuscript:

We look forward to receiving your revised manuscript.

Kind regards,

Iratxe Puebla

Senior Managing Editor, PLOS ONE

Journal Requirements:

I have read the journal's policy and the authors of this manuscript have the following competing interests: NSB is a paid employee of PT Biofarma. PT Biofarma is a state-owned company (owned by Indonesia government) that provides all vaccines for Indonesia National Immunization Program. Additionally, PT Biofarma was one of the funders for the RV3-BB Phase IIB trial. The trial was a phase IIB trial and while the RV3-BB rotavirus vaccine is not yet marketed, this vaccine is being developed with the intention for eventual usage in the national immunization program. This does not alter our adherence to PLOS ONE policies on sharing data and materials.

We note that you received funding from a commercial source: PT Biofarma.

Reviewers' comments:

Reviewer's Responses to Questions

**Comments to the Author**

1. Is the manuscript technically sound, and do the data support the conclusions?

Reviewer #1: Yes

Reviewer #2: Yes

2. Has the statistical analysis been performed appropriately and rigorously? 

Reviewer #1: Yes

Reviewer #2: Yes

3. Have the authors made all data underlying the findings in their manuscript fully available?

Reviewer #1: Yes

Reviewer #2: Yes

4. Is the manuscript presented in an intelligible fashion and written in standard English?

Reviewer #1: Yes

Reviewer #2: Yes

5. Review Comments to the Author

Reviewer #1: Table 5: column 5 does not show % of participants

Table 6: whats the difference between other infections disease group and other viral infections.Are viral infections included in other infections.I think it can be combined to one group as other infections

Reviewer #2: There are two points to be modified in the coming revision:

1. Reading through the manuscript, I found only tables presented in a comprehensive way but no any figures demonstrated. I would recommend the authors present their results in Figures (e.g., pie charts).

2. Regarding the statistical assessments on the data collected, it is clear that there are only very descriptive metrics calculated while the more comparative statistics should be performed to get insights of the cohorts. It is also necessary to include more sophisticated statistics for a report on cohort studies.

6. PLOS authors have the option to publish the peer review history of their article (what does this mean?). If published, this will include your full peer review and any attached files.

Reviewer #1: No

Reviewer #2: Yes: Xi-Nian Zuo

---

## [Author Response · Author response to Decision Letter 0]

31 Dec 2019

Dear Academic Editor and Reviewers

Thank you for spending your valuable time to read our previous manuscript and giving insightful suggestions to help us improve the quality of our manuscript.

Along with this letter are the Academic Editor and Reviewers comments, followed by our point-by-point response in addition to making the changes in the revised manuscript. Responses to the Academic Editor and Reviewers’ comments are typed in blue font, and the changes made in the manuscript are marked using track changes. Each author has given approval to the final form of this revision.

In this cover letter, we also would like to propose the following updated competing interests statement as suggested by the Academic Editor:

“The authors have the following interests: NSB is a paid employee of PT Biofarma. PT Biofarma is a state-owned company (owned by Indonesia government) that provides all vaccines for Indonesia National Immunization Program. Additionally, PT Biofarma was one of the non-commercial funders for the RV3-BB Phase IIB trial. No consultation was done to the funder relating to the trial. The trial was a phase IIB trial and while the RV3-BB rotavirus vaccine is not yet marketed, this vaccine is being developed with the intention for eventual usage in the national immunization program. This does not alter our adherence to PLOS ONE policies on sharing data and materials.”

We believe that the revision on our manuscript will improve the quality of our paper. We look forward to hearing from you in due course.

Sincerely,

Jarir At Thobari, MD, DPharm, PhD

Department of Pharmacology and Therapy and Pediatric Research Office (PRO) Faculty of Medicine, Public Health, and Nursing

Universitas Gadjah Mada

---

## [Decision Letter · Decision Letter 1]

10 Jun 2020

PONE-D-19-26757R1

Non-antibiotic Medication Use in an Indonesian Community Cohort 0-18 Months of Age

PLOS ONE

Dear Dr. At Thobari,

Thank you for submitting your manuscript to PLOS ONE. After careful consideration, we feel that it has merit but does not fully meet PLOS ONE’s publication criteria as it currently stands. Therefore, we invite you to submit a revised version of the manuscript that addresses the points raised during the review process.

We look forward to receiving your revised manuscript.

Kind regards,

Natasha McDonald

Associate Editor

PLOS ONE

Reviewers' comments:

Reviewer's Responses to Questions

**Comments to the Author**

1. If the authors have adequately addressed your comments raised in a previous round of review and you feel that this manuscript is now acceptable for publication, you may indicate that here to bypass the “Comments to the Author” section, enter your conflict of interest statement in the “Confidential to Editor” section, and submit your "Accept" recommendation.

Reviewer #1: All comments have been addressed

Reviewer #3: (No Response)

2. Is the manuscript technically sound, and do the data support the conclusions?

Reviewer #1: Yes

Reviewer #3: Partly

3. Has the statistical analysis been performed appropriately and rigorously? 

Reviewer #1: Yes

Reviewer #3: No

4. Have the authors made all data underlying the findings in their manuscript fully available?

Reviewer #1: Yes

Reviewer #3: Yes

5. Is the manuscript presented in an intelligible fashion and written in standard English?

Reviewer #1: Yes

Reviewer #3: Yes

6. Review Comments to the Author

Reviewer #1: one additional comment-How statistics were done for table 4.Please elaborate as its not easy to understand

Reviewer #3: Abstract: Early on in the abstract make the age range of participants clear.

What was the original aim of the study before the concomitant meds were looked at in this paper?

Add precision around estimates.

Line 207: make it clear what the numbers are (presume standard deviation).

The regression: how does this add to your clinical understanding?

What is the distribution of the variable number of drug regimens? Is linear regression the best model?

Are the assumptions met?

Was patient included as a variable in the model? Is it modelling how many drug regimens at a given time (or age) or overall from 0-18 months in a child’s life? This is a little unclear.

Table 7 include precision estimates.

How is 1518 people 100% when there were over 1600 patients on study? Did the others not have AES, or just not have it recorded? This is an important distinction to be made.

Line 300. Add a .

Line 389 and 390. How big were these 2 studies? If they were small there may not be a significant difference in the rates.

Figure 2 and 3- each patient could contribute multiple times to each result couldn’t they?

7. PLOS authors have the option to publish the peer review history of their article (what does this mean?). If published, this will include your full peer review and any attached files.

Reviewer #1: No

Reviewer #3: No

---

## [Author Response · Author response to Decision Letter 1]

24 Jun 2020

Dear Academic Editor and Reviewers

Thank you for spending your valuable time to read our previously revised manuscript and keep giving insightful suggestions to help us improve the quality of our manuscript.

Along with this letter are the Reviewers comments, followed by our point-by-point response in addition to making the changes in the revised manuscript. Responses to the Reviewers’ comments are typed in blue font, and the changes made in the manuscript are marked using track changes. Each author has given approval to the final form of this revision.

We believe that the revision on our manuscript will improve the quality of our paper. We look forward to hearing from you in due course.

Sincerely,

Jarir At Thobari, MD, DPharm, PhD

Department of Pharmacology and Therapy and Pediatric Research Office (PRO) 

Faculty of Medicine, Public Health, and Nursing

Universitas Gadjah Mada

1. Reviewer #1: one additional comment-How statistics were done for table 4.Please elaborate as its not easy to understand

Response: We thank the reviewer for noting this issue. After re-considering the best model to explore the factors that influenced the number of drug regimen in our study, we decided that instead of using a linear regression as in our previous manuscript, a logistic regression is better to provide the RR. Therefore, at the following lines we have added some changes:

Line 230-238: “A logistic regression was performed to explore the relationship between the reason of medication use, drug formulation, outpatient/inpatient set-ting, and the age of participant when using the medication with the number of non-antibiotic drug regimen.“

We also have revised the Table 4 and lines 348-355 (discussion section).

2. Reviewer #3: Abstract: Early on in the abstract make the age range of partici-pants clear.

Response: Thank you for noting this, now we have added “aged 0-18 months” at line 53

“A post-hoc study of the RV3-BB Phase IIb trial to children aged 0-18 months which was conducted in Indonesia during January 2013 to July 2016. Any concomitant medication use and health events among 1621 trial participants during the 18 months of follow-up were documented. Information on medication use included the frequency, formulation, indication, duration of usage, number of regimens, medication types, and therapeutic classes.”

3. What was the original aim of the study before the concomitant meds were looked at in this paper?

Response: The original aim of the study was primarily to evaluate the efficacy of RV3-BB vaccine against severe rotavirus gastroenteritis in children aged up to 18 months. This explanation has been stated at line 100-102.

4. Add precision around estimates.

Line 207: make it clear what the numbers are (presume standard deviation).

Response: Thank you for your suggestion. Now we have added some changes:

Line 205: with an average of 5.69 ± 4.80 drugs/child.

Line 207: Average use of non-antibiotic medications per AE was 1.72 (SD±0.99). 

The regression: how does this add to your clinical understanding?

What is the distribution of the variable number of drug regimens? Is linear re-gression the best model? Are the assumptions met? Was patient included as a variable in the model? Is it modelling how many drug regimens at a given time (or age) or overall from 0-18 months in a child’s life? This is a little unclear.

Response: We thank the reviewer for noting this issue. After re-considering the best model to explore the factors that influenced the number of drug regimen in our study, we decided that instead of using a linear regression as in our previous manuscript, a logistic regression is better to provide the RR. Therefore, at the following lines we have added some changes:

Line 230-238: “A logistic regression was performed to explore the relationship between the reason of medication use, drug formulation, outpatient/inpatient set-ting, and the age of participant when using the medication with the number of non-antibiotic drug regimen.“

We also have revised the Table 4 and lines 348-355 (discussion section).

5. Table 7 include precision estimates.

How is 1518 people 100% when there were over 1600 patients on study? Did the others not have AES, or just not have it recorded? This is an important dis-tinction to be made.

Response: Thank you for noting this. At table 7, we only included participants who had at least one AE during the study, therefore 1518 (of 1621) was 100%. Now we have added this information under the table.

“The table included only participants who had at least one AE during the 18 months of follow-up (N=1518/1621). The list is ordered by the proportion of ad-verse event (AE) episodes treated with non-antibiotic medications. SE: Standard Error of proportion.”

6. Line 300. Add a .

Response: Thank you for noting this, we have added . at line 300 (now at line 301).

7. Line 389 and 390. How big were these 2 studies? If they were small there may not be a significant difference in the rates.

Response:

Thank you for your suggestion, now we have revised the articles at line 390-393 by adding the total children of each study:

“Half of our participants used at least one systemic antihistamine during their first 18 month of life, which is higher compared to other studies showing 31% of 4511 children had used antihistamines by the age of 2 years in the Netherlands [1], and 17% of 1701 children during the first 5 years of life in Sweden [20].”

8. Figure 2 and 3- each patient could contribute multiple times to each result couldn’t they?

Response: Yes, each patient could contribute multiple times to each result. Now we have added this information under the figure legend. Thank you.

---

## [Decision Letter · Decision Letter 2]

11 Sep 2020

PONE-D-19-26757R2

Non-antibiotic Medication Use in an Indonesian Community Cohort 0-18 Months of Age

PLOS ONE

Dear Dr. At Thobari,

Thank you for submitting your manuscript to PLOS ONE. After careful consideration, we feel that it has merit but does not fully meet PLOS ONE’s publication criteria as it currently stands. Therefore, we invite you to submit a revised version of the manuscript that addresses the points raised during the review process.

We look forward to receiving your revised manuscript.

Kind regards,

Emma Link, DPhil

Academic Editor

PLOS ONE

Additional Editor Comments (if provided):

I can see some alterations to the paper, which assist with clarity of understanding. There are a few other areas for consideration before this could be published.

In the statistical analysis section line 185 of track changed doc it states that "descriptive data, the results were presented as mean, median, frequency and percentages". Then on line 207 it states "an average of 5.69 +/- 4.80 drugs/child". Is the 4.80 the median? If not, please explain what this is, and/or change the statistical analysis methods.

Reviewers' comments:

Reviewer's Responses to Questions

**Comments to the Author**

1. If the authors have adequately addressed your comments raised in a previous round of review and you feel that this manuscript is now acceptable for publication, you may indicate that here to bypass the “Comments to the Author” section, enter your conflict of interest statement in the “Confidential to Editor” section, and submit your "Accept" recommendation.

Reviewer #1: (No Response)

2. Is the manuscript technically sound, and do the data support the conclusions?

Reviewer #1: Yes

3. Has the statistical analysis been performed appropriately and rigorously? 

Reviewer #1: Yes

4. Have the authors made all data underlying the findings in their manuscript fully available?

Reviewer #1: Yes

5. Is the manuscript presented in an intelligible fashion and written in standard English?

Reviewer #1: Yes

6. Review Comments to the Author

Reviewer #1: 1.Table 5: Row 5 and row 14- what does antiinfective agents in row 5 mean? is it antibiotics for GI infections .If it is antibiotics then it should be removed as table is showing non antibiotic medications. Similarly I believe antiseptics and disinfectants in row 15 means topical agents

2.Table 5 : row 7 " drugs for obstructive airway disease " I believe are bronchodilators. Bronchodilators gives a better understanding for these drugs

Table 5: row 8 -are these corticosteroids for topical use only?

3.Figure 3:Ear disorders - antiseptics and disinfectants are topical preparations?

4.Line 343-344: reformatting of sentence as " This also explains a usage/consumption of higher number of drugs by older infants...something like that

5.Line 345-346.Reformatting of sentence The inclusion of prescription and non prescription drugs and the exclusion of antibiotics in our study.....

6.Lline 352,353.reformatting of sentence.Respiratory disorders,gastrointestinal disorders and unspecified pyrexia manifest as multiple symptoms....

7.Line 364.sentence formatting.Similar to other studies,respirator system disorder was the primary indication for medication use in our study

8.Line 371-373.sentence formatting.analgesics/antipyretics....sentence is not very clear.This drug usage pattern was similar to other studies

9.Line 378s-seems like you are comparing the same age group population.it does not seem different in age.Please clarify

10.Line386-whats CCMs

11.Line 396..sentence formatting..higher incidence of respiratory tract infections

12Line 408-409.Its a better idea to mention percent of deaths from each drug

13Line 427.sentence formatting.Although not being investigated in children..

14.conclusion paragraph reformatting.The majority of children were exposed.....A variety of medications were used.....This calls for further education....

7. PLOS authors have the option to publish the peer review history of their article (what does this mean?). If published, this will include your full peer review and any attached files.

Reviewer #1: No

---

## [Author Response · Author response to Decision Letter 2]

7 Oct 2020

I. Academic Editor comment 

1. I can see some alterations to the paper, which assist with clarity of understanding. There are a few other areas for consideration before this could be published.

In the statistical analysis section line 185 of track changed doc it states that "descriptive data, the results were presented as mean, median, frequency and percentages". Then on line 207 it states, "an average of 5.69 +/- 4.80 drugs/child". Is the 4.80 the median? If not, please explain what this is, and/or change the statistical analysis methods.

Response: Thank you for your suggestion. The 4.80 is a standard deviation of the mean (5.69). We have revised the “average” on line 205 into “a mean (SD)”.

II. Reviewer #1 comments 

1. Table 5: Row 5 and row 14- what does anti-infective agents in row 5 mean? is it anti-biotics for GI infections. If it is antibiotics then it should be removed as table is showing non antibiotic medications. Similarly, I believe antiseptics and disinfectants in row 15 means topical agents

Response: We apologize for the ambiguity in table %. In Table 5 Row 5, we used the exact ATC WHO therapeutic class name of “antidiarrheals, intestinal antiinflammary /anti-infective agents”. However, we have already removed the anti-infective drugs from the very beginning. Therefore, to avoid this ambiguity, now we have revised the therapeutic class name in Table 5 Row 5 as “antidiarrheals and intestinal antiinflammary agents” only. 

In Table 5, Row 14, indeed, antiseptics and disinfectants are topical agents. Now we have added such information in Table 5 Row 14. This additional information is also for addressing the reviewer 3rd comment “ear disorders – antiseptics and disinfectants are topical preparations only?”

2. Table 5: Row 7 “drugs for obstructive airway disease” I believe are bronchodilators. Bronchodilators gives a better understanding for these drugs.

Table 5: Row 8 –are these corticosteroids for topical use only?

Response: Row 7: Yes, this therapeutic class of ATC WHO contains any bronchodilator types. We have revised it into “Bronchodilators” as your suggestion. Thank you.

In Row 8 – Yes, the corticosteroids for dermatological preparations are topical agents as well. Now we have revised it into “corticosteroids (topical agents)”. Thank you for noting this.

3. Figure 3: Ear disorders – antiseptics and disinfectants are topical preparations?

Response: We have added an additional information to address this suggestion on Table 5 Row 14.

4. Line 343-344: reformatting of sentence as " This also explains a usage/consumption of higher number of drugs by older infants...something like that

Response: Thank you. We have reformatted the sentence as your suggestion on line 342-343: “This also explains a consumption of higher number of drugs by older infants (Table 4).”

5. Line 345-346.Reformatting of sentence the inclusion of prescription and non-prescription drugs and the exclusion of antibiotics in our study.....

Response: Thank you. We have reformatted the sentence as your suggestion on line 344: “The inclusion of prescription and non-prescription drugs and the exclusion of antibiotics in our study also might affect the average number of drug regimen compared to those who calculated prescribed drugs only without excluding antibiotic, as non-prescribed drug tend to be used in milder conditions that need fewer number of drug per regimen.”

6. Line 352,353. reformatting of sentence. Respiratory disorders, gastrointestinal disorders and unspecified pyrexia manifest as multiple symptoms....

Response: Thank you. We have reformatted the sentence as your suggestion on line 351: “Respiratory disorders, gastrointestinal disorders, and unspecified pyrexia manifest as multiple symptoms, thus may encourage child caretakers to self-medicate the children with multiple over-the-counter (OTC)”

7. Line 364.sentence formatting. Similar to other studies, respirator system disorder was the primary indication for medication use in our study

Response: Thank you. We have reformatted the sentence as your suggestion on line 361-362: “Similar to other studies, respiratory system disorder was the primary indication for medication use in our study”.

8. Line 371-373.sentence formatting Analgesics/antipyretics....sentence is not very clear. This drug usage pattern was similar to other studies

Response: Thank you. We have reformatted the sentence as your suggestion on line 370: “This drug usage pattern was similar to other studies”

9. Line 378s-seems like you are comparing the same age group population.it does not seem different in age. Please clarify

Response: The age of our participants ranged from 0-18 months, compared to the age of 0-18 years in the other study. This might sound similar and confuse the reader, therefore we have reformatted the sentence in line 375 into: “The reasons for different results were difficult to determine, but possible explanations might be due to the younger age of participants in our study.” 

10. Line 386-what CCMs

Response: Thank you for noting this. CCMs is an abbreviation of cough and cold medications. We have added this information in the first abbreviation in line 384.

11. Line 396..sentence formatting Higher incidence of respiratory tract infections

Response: Thank you. We have reformatted the sentence in line 393-394 as your suggestion into “This might partly reflect a higher incidence of respiratory tract infections in our population.” 

12. Line 408-409.Its a better idea to mention percent of deaths from each drug

Response: Thank you for your suggestion. We have put the percentage of deaths from each drug in line 405-406.

13. Line 427.sentence formatting Although not being investigated in children.

Response: Thank you. We have reformatted the sentence in line 424 as your suggestion into: “Although not being investigated in children, vitamin B6 overdose in adult has been known to induce neurotoxic syndrome”.

14. Conclusion paragraph reformatting The majority of children were exposed.....A variety of medications were used.....This calls for further education....

Response: Thank you. We have reformatted the sentence in conclusion paragraph (line 442) as your suggestion into: “The majority of children were exposed to at least one medication during their early life. A variety of medications were used, including those where the evidence of efficacy and safety in pediatric population is lacking. This calls for further education to improve caregivers’ knowledge on appropriate self-medication in children, and to enhance the guideline compliance by health practitioners to ensure the therapy given to children will be of most benefit.“

---

## [Decision Letter · Decision Letter 3]

3 Nov 2020

Non-antibiotic Medication Use in an Indonesian Community Cohort 0-18 Months of Age

PONE-D-19-26757R3

Dear Dr. At Thobari,

We’re pleased to inform you that your manuscript has been judged scientifically suitable for publication and will be formally accepted for publication once it meets all outstanding technical requirements.

Kind regards,

Emma Link, DPhil

Guest Editor

PLOS ONE

Additional Editor Comments (optional):

I participated as a reviewer for the initial evaluation of this manuscript.

Reviewers' comments:

Reviewer's Responses to Questions

**Comments to the Author**

1. If the authors have adequately addressed your comments raised in a previous round of review and you feel that this manuscript is now acceptable for publication, you may indicate that here to bypass the “Comments to the Author” section, enter your conflict of interest statement in the “Confidential to Editor” section, and submit your "Accept" recommendation.

Reviewer #1: All comments have been addressed

2. Is the manuscript technically sound, and do the data support the conclusions?

Reviewer #1: Yes

3. Has the statistical analysis been performed appropriately and rigorously? 

Reviewer #1: Yes

4. Have the authors made all data underlying the findings in their manuscript fully available?

Reviewer #1: Yes

5. Is the manuscript presented in an intelligible fashion and written in standard English?

Reviewer #1: Yes

6. Review Comments to the Author

Reviewer #1: (No Response)

7. PLOS authors have the option to publish the peer review history of their article (what does this mean?). If published, this will include your full peer review and any attached files.

Reviewer #1: No

---

## [Editor Report · Acceptance letter]

9 Nov 2020

PONE-D-19-26757R3 

Non-antibiotic Medication Use in an Indonesian Community Cohort 0-18 Months of Age 

Dear Dr. At Thobari:

I'm pleased to inform you that your manuscript has been deemed suitable for publication in PLOS ONE. Congratulations! Your manuscript is now with our production department. 

Kind regards, 

on behalf of

Dr. Emma Link 

Guest Editor

PLOS ONE